# T2VPhysBench: A First-Principles Benchmark for Physical Consistency in Text-to-Video Generation

## Abstract

Text-to-video generative models have made significant strides in recent years, producing high-quality videos that excel in both aesthetic appeal and accurate instruction following, and have become central to digital art creation and user engagement online. Yet, despite these advancements, their ability to respect fundamental physical laws remains largely untested: many outputs still violate basic constraints such as rigid-body collisions, energy conservation, and gravitational dynamics, resulting in unrealistic or even misleading content. Existing physical-evaluation benchmarks typically rely on automatic, pixel-level metrics applied to simplistic, life-scenario prompts, and thus overlook both human judgment and first-principles physics. To fill this gap, we introduce **T2VPhysBench**, a first-principled benchmark that systematically evaluates whether state-of-the-art text-to-video systems, both open-source and commercial, obey twelve core physical laws including Newtonian mechanics, conservation principles, and phenomenological effects. Our benchmark employs a rigorous human evaluation protocol and includes three targeted studies: (1) an overall compliance assessment showing that all models score below 0.60 on average in each law category; (2) a prompt-hint ablation revealing that even detailed, law-specific hints fail to remedy physics violations; and (3) a counterfactual robustness test demonstrating that models often generate videos that explicitly break physical rules when so instructed. The results expose persistent limitations in current architectures and offer concrete insights for guiding future research toward truly physics-aware video generation.

## 1 Introduction

Text-to-video generative models (Singer et al., 2023; Wu et al., 2023; Hong et al., 2023; Yang et al., 2024b) have achieved remarkable success in recent years, driven by advances in the Transformer architecture (Arnab et al., 2021; Liu et al., 2022) and diffusion model techniques (Ho et al., 2022b; Esser et al., 2023). By leveraging large-scale, cross-modal video–text data from the Internet, these models now produce videos with high fidelity and appealing aesthetics, transforming both digital art creation and user engagement on the Web. Modern systems such as Sora (OpenAI, 2024), WanX (Alibaba, 2025) and Kling (Kling, 2024) have demonstrated the ability to follow complex human instructions with impressive accuracy, positioning text-to-video generation as a central feature of today's web experiences.

Despite these gains, fundamental concerns remain about whether text-to-video models respect basic physical laws (Lv et al., 2024; Lin et al., 2025; Motamed et al., 2025; Wang et al., 2025). Generated videos often violate constraints such as rigid-body collisions, fluid dynamics, or simple gravity, which can lead to unrealistic or even misleading content. Such errors become critical in applications like robotics (Yang et al., 2024a; Du et al., 2023) and autonomous driving (Santana & Hotz, 2016; Zhou et al., 2024; Wen et al., 2024), where adherence to real-world physics is essential for safety and system reliability. It is therefore crucial to evaluate how well current models capture these core principles.

Recent years have seen a growing suite of benchmarks for text-to-video models, covering compositional property combinations (Sun et al., 2024; Li et al., 2024), temporal dynamics (Ji et al., 2024;

Liao et al., 2024), object counting (Guo et al., 2025) and storytelling (Bugliarello et al., 2023). However, systematic evaluation of physical constraint adherence remains underexplored. Early, pioneering benchmarking efforts on this topic have introduced physics-inspired prompts and provided valuable insights, but they typically rely on pixel-level or visual-matching metrics that do not fully align with human judgments (Lin et al., 2025). In addition, most existing tests use scenario-based designs rather than grounding tasks in first-principles laws (e.g., Newton's laws or Bernoulli's principle) (Wang et al., 2025; Meng et al., 2024a; Motamed et al., 2025). To bridge these gaps, a human-centered, law-driven benchmark is needed to more faithfully reflect real-world physical understanding and to guide future improvements.

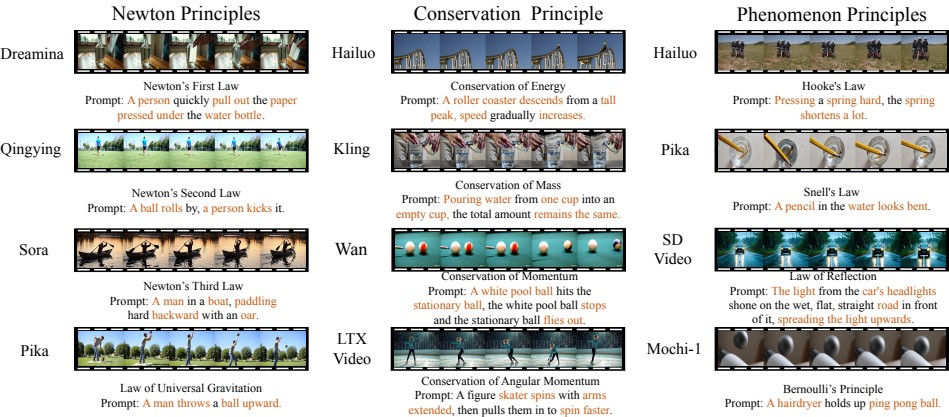

Figure 1: **All 12 physical laws evaluated in this benchmark, illustrated with video examples from various text-to-video models.**

In this paper, we introduced a human-evaluated, first-principles benchmark, namely **T2VPhysBench**, designed to assess whether text-to-video models can follow 12 fundamental physical laws (see Figure 1). We include both leading open-source models and state-of-the-art commercial systems, reflecting the latest advances in 2025. Our study exposes persistent challenges in modeling physical behavior and offers insights into why models fail.

Our contributions can be summarized as follows:

- We introduce a first first-principled benchmark that systematically evaluates whether modern text-to-video generation models respect twelve fundamental physical laws, covering Newtonian mechanics, conservation principles, and phenomenological effects.

- Through a rigorous human evaluation protocol, we demonstrate that all state-of-the-art text-to-video models consistently fail to satisfy even basic physical constraints, with average compliance scores below 0.60 across every law category.

- By incorporating progressively more concrete hints, naming the law and adding detailed mechanistic descriptions, we show that prompt refinement alone cannot overcome the models' inability to generate physically coherent videos.

- We challenge models with counterfactual prompts that explicitly request physically impossible scenarios and find that they often comply, producing rule-violating videos and revealing a reliance on surface patterns rather than true physical reasoning.

**Roadmap.** In Section 2, we review some prior works. In Section 3, we show the details of our proposed benchmark. In Section 4, we present the main evaluation results with our benchmark. In Section 5, we provide some insights to understand the failure of text-to-video models in following physical constraints. In Section 6, we draw a conclusion for this paper.

## 2 RELATED WORKS

**Benchmarks on Text-to-Video Generation.** As text-to-video models have been a fundamental game changer in users' online experiences, particularly in creative art creation, their evaluation

has become a crucial area of focus. Existing benchmarks on text-to-video models have covered a wide range of aspects, including basic video fidelity (Liu et al., 2023), the compositional ability of different keywords (Sun et al., 2024; Feng et al., 2024), temporal dynamics (Ji et al., 2024; Liao et al., 2024), and complex storytelling capabilities (Bugliarello et al., 2023). Benchmarking how text-to-video generation models adhere to basic physical laws is another key area of evaluation (Meng et al., 2024a; Motamed et al., 2025; Bansal et al., 2025; Meng et al., 2024b; Wang et al., 2025). For example, VideoPhy (Bansal et al., 2025) proposes a human-evaluated benchmark that systematically examines collisions between different materials, such as solid-solid, solid-fluid, and fluid-fluid cases. The Physics-IQ benchmark (Motamed et al., 2025) evaluates models based on their ability to extend given video frames, assessing the extended frames using automated evaluation metrics like MSE or IoU. While these works provide valuable early insights into evaluating the physical behavior of text-to-video models, they do not approach the problem from a first-principles physical law perspective, nor do they incorporate careful human evaluation, which highlights the need for our work.

**Text-to-Video Generative Models.**  Text-to-video has long been a central topic in generative AI. Early approaches to text-to-video can be traced back to VAEs (Kingma & Welling, 2022; Li et al., 2018) and GANs (Pan et al., 2017; Goodfellow et al., 2020; Balaji et al., 2019) conditioned on text, which were limited by the weak generative abilities of early models and the weak connection between video and text. Empowered by large-scale visual-text pretraining (Radford et al., 2021; Xu et al., 2021; Li et al., 2022) with vast amounts of data from the Internet, and the development of modern video diffusion models (Ho et al., 2022b; Harvey et al., 2022; Blattmann et al., 2023b), recent text-to-video generative models (Ho et al., 2022a; Wang et al., 2023; Singer et al., 2023; Yang et al., 2024b; Zhang et al., 2024) have significantly improved the quality of generated videos and their ability to follow complex textual prompts. For instance, Imagen Video (Ho et al., 2022a) builds on prior work in diffusion-based image generation, extending it to video through a cascade of spatial and temporal super-resolution models, progressive distillation, and classifier-free guidance for improved fidelity and control. Similarly, Make-A-Video extends text-to-image generation to text-to-video (Singer et al., 2023) by integrating spatial-temporal modules, a decomposed temporal U-Net, and a multi-stage pipeline with super-resolution models, enabling high-quality video synthesis without paired text-video data. Despite the strong video fidelity and instruction-following abilities of these text-to-video diffusion models, their fundamental capability to adhere to simple physical laws still exhibits significant gaps (Lv et al., 2024; Meng et al., 2024a; Lin et al., 2025), which is one of the key motivations for this benchmark.

## 3 THE T2VPHYSBENCH BENCHMARK

In this section, we first present the baseline video generation models in Section 3.1, then introduce our benchmark prompts in Section 3.2, and finally describe the evaluation protocol in Section 3.3.

### 3.1 BASELINE MODELS

Table 1: **Key information of the 10 text-to-video models in this benchmark.**

| Model Name | Year | # Params | Organization | Open |
|:---:|:---:|:---:|:---:|:---:|
| Kling (Kling, 2024) | 2024 | N/A | Kuai | No |
| Wan 2.1 (Alibaba, 2025) | 2025 | 14B | Alibaba | Yes |
| Sora (OpenAI, 2024) | 2024 | N/A | OpenAI | No |
| Mochi-1 (Genmo, 2024) | 2024 | 10B | Genmo | Yes |
| LTX Video (HaCohen et al., 2024) | 2024 | 2B | Lightricks | Yes |
| Pika 2.2 (Pika, 2024) | 2025 | N/A | Pika Labs | No |
| Dreamina (ByteDance, 2024) | 2024 | N/A | ByteDance | No |
| Qingying (Zhipu, 2024) | 2024 | 5B | Zhipu | Yes |
| SD Video (Blattmann et al., 2023a) | 2023 | 1.4B | Stability AI | Yes |
| Hailuo (MiniMax, 2025) | 2025 | N/A | MiniMax | No |

We selected a diverse set of state-of-the-art video generation models released between 2023 and 2025 to ensure our evaluation reflects the latest advances and uncovers their limitations in following

physical constraints. Our benchmark includes ten models, spanning both closed-source and open-source systems. Detailed model specifications are listed in Table 1.

For generation, we use the lowest available resolution (typically 720p) to balance visual fidelity with physical accuracy. We fix a 16:9 aspect ratio and choose a short video duration (usually 4 seconds) to concentrate the evaluation on fundamental physical behaviors. Further implementation details are provided in Appendix A.

## 3.2 BENCHMARK PROMPTS

In this benchmark, we address the problem of enforcing physical constraints using a first-principles approach. Rather than relying on intuition or everyday contexts, our prompts are derived directly from fundamental laws of physics. We organize these laws into three categories: Newton's laws, conservation laws, and phenomenological principles. In each category we select four specific laws (for a total of twelve), and for each law we design seven prompts based on realistic scenarios. Consequently, each model is evaluated on 84 distinct prompts. Example prompts and their video outputs are presented in Figure 1.

**Newton's Principles.** This category comprises Newton's three laws of motion and the law of universal gravitation. For the first law (inertia), we consider objects in free space or under no net external force, which should remain at rest or move at constant velocity. For the second law (force–acceleration relation), we apply a known external force to a specified object (e.g., a person pushing a box) and verify that the resulting acceleration matches the prediction. For the third law (action–reaction), we examine interactions in a fluid or gas medium, for example, pushing an object backward or downward and observing the equal and opposite response. Finally, for the law of universal gravitation, prompts include tossing an object in Earth's gravitational field or depicting planets orbiting under mutual attraction.

**Conservation Principles.** This category includes four fundamental conservation laws: conservation of energy, mass, linear momentum, and angular momentum. For the conservation of energy, we design prompts that involve conversions between potential and kinetic energy, such as a roller coaster descending, two colliding balls exchanging motion, or a compressed spring releasing its stored energy. For conservation of mass, we consider scenarios where matter changes form but not quantity, including melting ice in a sealed container or transferring liquids between containers while keeping the total mass unchanged. Conservation of linear momentum is explored through interactions like elastic collisions between carts, or a person throwing a heavy object and recoiling on a skateboard, demonstrating momentum transfer in closed systems. Finally, conservation of angular momentum is illustrated using rotating systems such as figure skaters pulling in their arms to spin faster, or individuals shifting their position on a spinning platform to change the rotation rate.

**Phenomenon Principles.** This category consists of physical laws that describe specific observable effects, such as Hooke's Law, Snell's Law, the Law of Reflection, and Bernoulli's Principle. For Hooke's Law, we present prompts involving springs under varying forces to assess whether deformation is proportional to the applied force. Snell's Law is evaluated through optical distortions caused by refraction, such as the bending appearance of a pencil in water or the mirage-like effect of heat waves on a road. The Law of Reflection is tested by examples involving predictable angular deflections, including laser light hitting a metal surface or a ball rebounding off a wall. Finally, Bernoulli's Principle is represented through aerodynamic and fluid dynamic effects, such as air flowing around an airplane wing generating lift, or a hairdryer levitating a ping pong ball due to pressure differences.

## 3.3 EVALUATION PROTOCOL

To align with human judgment and address the fidelity-only limitations of prior physical benchmarks, we adopt a fully manual evaluation protocol, following VideoPhy (Bansal et al., 2025). Three annotators (undergraduate or graduate students) independently review every generated video and assign it one of four quality levels based on its adherence to the target physical law. Each level is then mapped to a real-valued score in $[0, 1]$:

- **Level 1** (score 0.0): the video fails to demonstrate the intended physical behavior.
- **Level 2** (score 0.25): the video exhibits a clear violation of the law.
- **Level 3** (score 0.5): the video is largely correct but contains minor inaccuracies.
- **Level 4** (score 1.0): the video fully and accurately conforms to the law.

This scoring scheme rewards fully correct generations while still allowing partial credit for near-correct cases. For each model, we average the scores across all prompts and annotators to produce a single physical-consistency score, which is then used to rank the models.

## 4 EXPERIMENTS

In this section, we show the main experiment results of our proposed benchmark. In Section 4.1, we present the observations from the overall result. In Section 4.2, we show the impact of different hint levels on the physical constraint following ability. In Section 4.3, we show how the text-to-video models perform under counterfactual prompts.

### 4.1 OVERALL PHYSICAL CONSTRAINT RESULTS

Table 2: **Score Across Different Principles.**

| Model | Newton Principles | Conservation Principles | Phenomenon Principles | Avg. Score |
|---|---|---|---|---|
| SD Video | 0.21 | 0.19 | 0.19 | 0.19 |
| Hailuo | 0.27 | 0.15 | 0.25 | 0.22 |
| Dreamina | 0.19 | 0.13 | 0.38 | 0.23 |
| Sora | 0.31 | 0.15 | 0.38 | 0.28 |
| LTX Video | 0.40 | 0.13 | 0.40 | 0.31 |
| Pika 2.2 | 0.38 | 0.19 | 0.40 | 0.32 |
| Mochi-1 | 0.40 | 0.23 | 0.40 | 0.34 |
| Kling | 0.52 | 0.17 | 0.38 | 0.35 |
| Qingying | 0.35 | 0.23 | 0.63 | 0.40 |
| Wan 2.1 | 0.56 | 0.29 | 0.42 | 0.42 |

We compare all the models listed in Table 1 and present the overall result in Table 2. Across all ten models, no system achieves even moderate accuracy on our proposed benchmark. First, the highest average score on Newton's principles is only 0.56 (Wan 2.1) and the lowest is 0.19 (Dreamina). Similarly, the best performance on conservation laws peaks at 0.29 (Wan 2.1), while the worst is just 0.13 (Dreamina and LTX Video). This indicates that current text-to-video models struggle to capture even the simplest physical behaviors.

**Observation 4.1.** *Despite advances in video generation, all evaluated models score below 0.60 on basic Newtonian and conservation laws, highlighting a consistent failure to model fundamental physics.*

Within each model, performance on conservation principles is consistently lower than on Newton's or phenomenon principles. For instance, LTX Video scores 0.13 on conservation but achieves 0.40 on Newton's laws and 0.40 on phenomenon principles. Similarly, Pika 2.2 attains 0.19 on conservation, yet scores 0.38 on Newton's principles and 0.40 on phenomenon principles. This pattern indicates that conservation laws pose a greater challenge, while models handle Newtonian dynamics and observable phenomena more successfully.

**Observation 4.2.** *The score variance between different types of laws is noticeable. Conservation principles are substantially harder for current models, whereas Newton's laws and phenomenon principles yield consistently higher scores.*

When comparing across models, the best overall performer (Wan 2.1) obtains an average score of 0.42, whereas the worst (SD Video) averages just 0.19, showing a gap of 0.23. Even among the top three, Mochi-1 and Kling achieve only 0.34 and 0.35, respectively. This large variance between different models strengthens the need for more robust physics grounding in future video-generation architectures.

**Observation 4.3.** *The difference between the highest and lowest average scores (0.42 vs. 0.19) reveals a substantial performance gap, motivating targeted improvements in physical reasoning capabilities.*

## 4.2 IMPACT OF HINT LEVELS

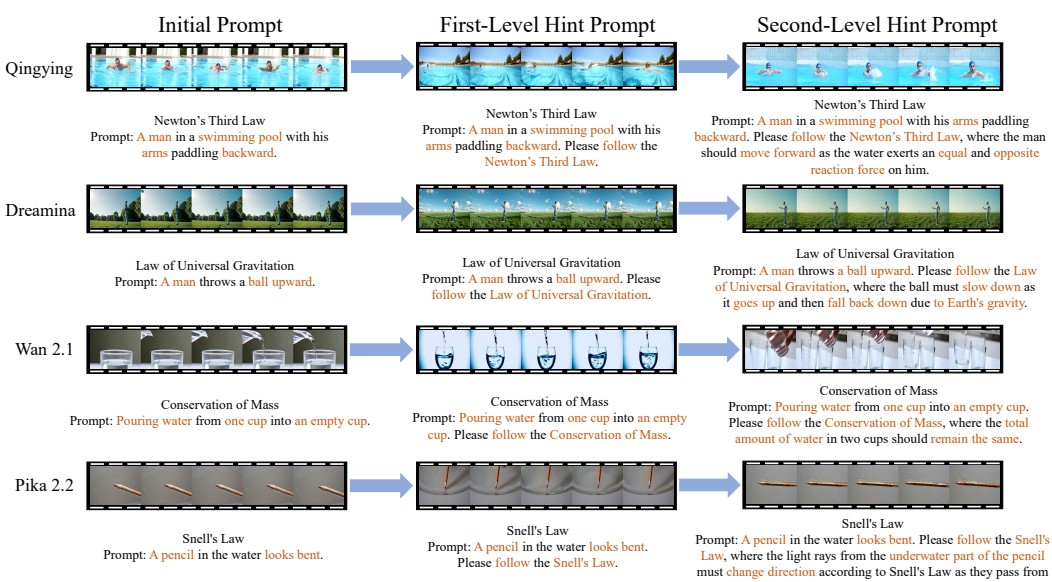

Figure 2: **Prompt and Video Examples with Different Hint Levels**.

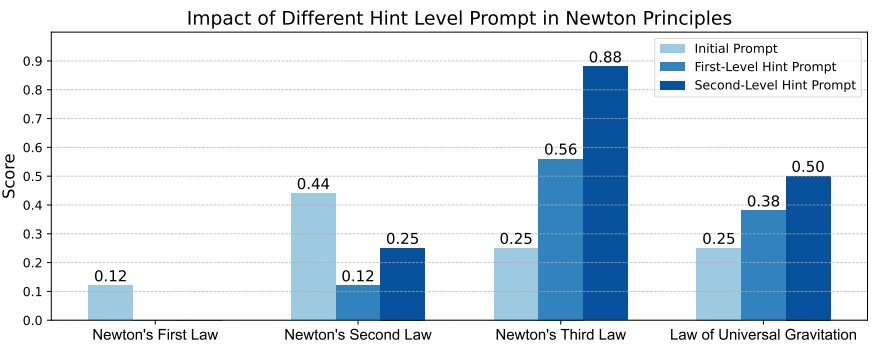

Figure 3: **Ablation Study of Different Hint Level Prompts in Newton Principles**.

Based on our previous findings in Section 4.1, we have observed that most text-to-video models fail to generate videos that comply with physical laws. To show that this inherent limitation is non-trivial and cannot be resolved simply through prompt improvements, in this study we explore a simple but critical problem: can text-to-video models follow physical constraints when provided with hints of different levels of concreteness? Specifically, we consider three hint levels (see Figure 2 for prompt and video examples), with details as follows:

- **Initial Prompt**: The original prompt without any additional hints. An example prompt could be: *"A spring is compressed and springs open when released."*
- **First-Level Hint**: The name of the relevant physical law is explicitly provided, simplifying the problem. An example prompt could be: *"A spring is compressed and springs open when released. Please follow the Conservation of Energy."*
- **Second-Level Hint**: A fully concrete scenario with detailed physical interpretation is provided, alongside naming the law. An example prompt could be: *"A spring is compressed*

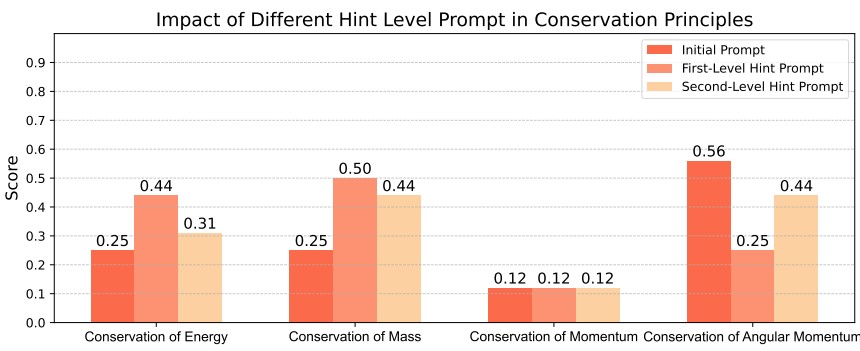

Figure 4: **Ablation Study of Different Hint Level Prompts in Conservation Principles**.

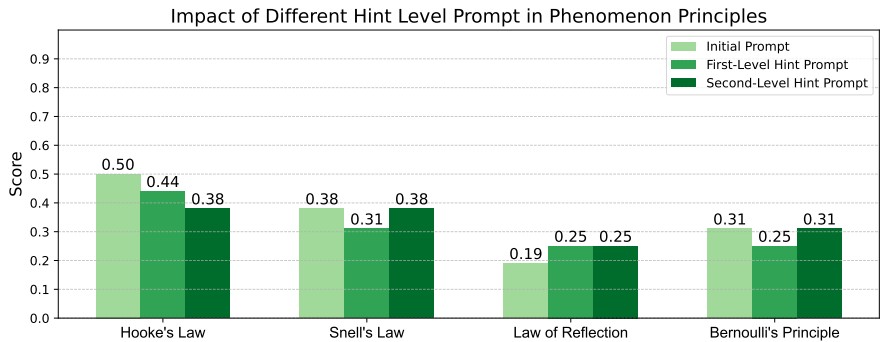

Figure 5: **Ablation Study of Different Hint Level Prompts in Phenomenon Principles**.

> *and springs open when released. Please follow the Conservation of Energy, where the potential energy stored in the compressed spring must be converted into kinetic energy upon release, ensuring the total energy remains constant."*

We present the experimental results on the impact of hint levels in Figure 3, Figure 4, and Figure 5, where the three different types of laws are shown independently. From the figures, the first observation is that despite some rare counterexamples, such as the consistent improvement of the average score from 0.25 to 0.88 on Newton's third law, and the improvement from 0.25 to 0.50 on the law of universal gravitation between two levels of hints, most physical laws do not exhibit significant improvement with enhanced hint levels.

More interestingly, prompt refinement through providing hints can even produce negative impacts. For instance, for Hooke's law, the score decreased from 0.50 to 0.38, and for Newton's first law, the score dropped from 0.12 to 0.00 even at the first level of hint. Such reductions sometimes occur only at the first hint level, as seen for Snell's law, where the initial prompt and second-level hint achieve a score of 0.38, but it reduces to 0.31 at the first level. In other scenarios, the score reduction occurs at both hint levels, such as in Hooke's law and Newton's second law. This leads to the following observation:

**Observation 4.4.** *Despite consistent improvements on a small number of physical laws, for most physical laws, increasing the hint level does not enhance the physical law-following scores, and in many cases, even leads to a negative impact at both hint levels.*

### 4.3 IMPACT OF COUNTERFACTUAL PROMPTS

To assess whether the models truly understand physical laws rather than rely on superficial pattern matching, we design counterfactual prompts that explicitly describe impossible scenarios. From a counterfactual perspective, a model with genuine physical reasoning should understand how to generate videos that violate some specific physical laws. For instance, an apple in a full vacuum

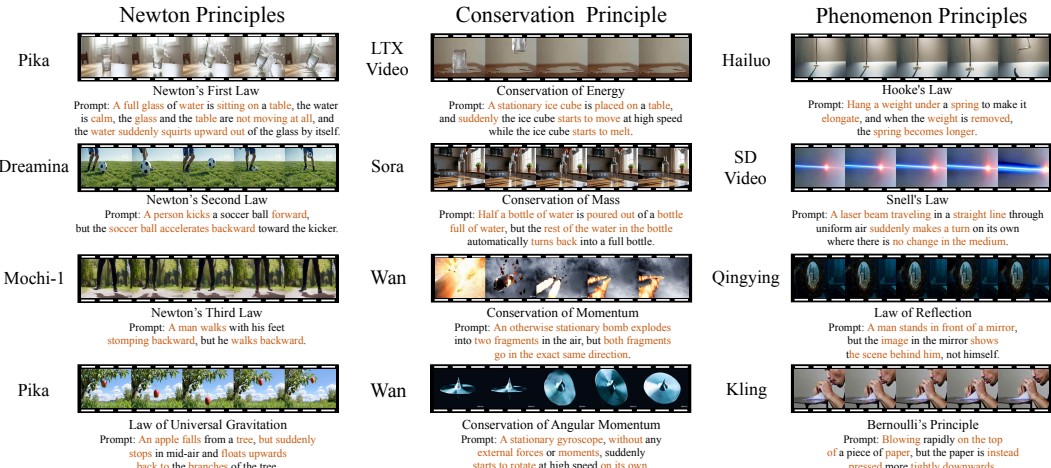

Figure 6: **Examples of counterfactual prompts in this benchmark, along with generated example videos from all ten text-to-video models.**

Table 3: **Impact of Counterfactual Prompts.**

| Model | Newton Principles | Conservation Principles | Phenomenon Principles | Avg. Score |
|-------|-------------------|-------------------------|------------------------|------------|
| Kling | 0.19 | 0.31 | 0.00 | 0.17 |
| Dreamina | 0.31 | 0.25 | 0.06 | 0.21 |
| Mochi-1 | 0.25 | 0.31 | 0.31 | 0.29 |
| Wan 2.1 | 0.31 | 0.50 | 0.13 | 0.31 |
| SD Video | 0.31 | 0.31 | 0.31 | 0.31 |
| LTX Video | 0.44 | 0.25 | 0.31 | 0.33 |
| Qingying | 0.50 | 0.38 | 0.31 | 0.40 |
| Sora | 0.38 | 0.56 | 0.44 | 0.46 |
| Hailuo | 0.31 | 0.63 | 0.44 | 0.46 |
| Pika 2.2 | 0.63 | 0.56 | 0.25 | 0.48 |

without any force (e.g., gravity) on it would be static or have uniform linear motions due to Newton's first law, while a counterfactual example that violates newton's first law should be an apple moving with some acceleration that is not uniform velocity. Example prompts and videos are presented in Figure 6.

Following a similar setting as the overall result in Table 4.1, we replace the original prompts with their counterfactual version, and then compute the average score for each principle type. The real-valued score has the same meaning as mentioned in Section 3.3. The results of this experiment is presented in Table 3.

From the experiment results, we can find that the scores remain uniformly low under these counterfactual conditions, since for all the scores, there are rare results that has better score than 0.50. For instance, Kling achieves only 0.19 on Newton's principles and even fail in all the prompts for the phenomenon principles, while Dreamina records a mere 0.06 on phenomenon principles. Therefore, we have the following observation.

**Observation 4.5.** *Even when instructed to violate the laws, all models score poorly in all the physical law classes, demonstrating an inability to understand impossible physics.*

Another noticeable finding is that high performance on the original prompts (see Table 4.1) does not necessarily translate to strong performance on counterfactual prompts. For example, Wan 2.1, originally the top performer with an average score of 0.42, falls to fourth-from-bottom (0.31) under counterfactual conditions, while SD Video, originally the worst at 0.19, rises to 0.31, matching Wan 2.1's counterfactual score. Even more revealing, Kling scored 0.38 on phenomenon principles in the standard evaluation but dropped to 0.00 on the counterfactual phenomenon prompts. These reversals indicate that apparent compliance under normal prompts arises from surface-level pattern matching rather than a true understanding of physical constraints.

**Observation 4.6.** *Models that excel under standard prompts can be easily misled by counterfactuals, showing their compliance is rooted in memorized patterns rather than genuine physical reasoning.*

## 5 DISCUSSION

In this section, we discuss several open directions and possible solutions to the inherent limitations of text-to-video models in adhering to physical constraints.

**Understanding and Prediction in World Foundation Models.** World Foundation Models (WFMs) refer to large neural networks that simulate physical environments and predict outcomes based on given inputs (Ha & Schmidhuber, 2018; Okada & Taniguchi, 2022; Agarwal et al., 2025). These models go beyond simple text and video matching, as seen in previous text-to-video models (Singer et al., 2023; Wu et al., 2023), by understanding the physical and spatial constraints of the real world. They possess the capability to make predictions using sensory data where motion, force, and spatial relationships are grounded in reality. When such a model is used as a backbone for video generation, the output naturally obeys learned physical rules. For instance, objects move consistently under acceleration, and interactions like pushing or stacking behave plausibly according to cause-and-effect principles.

**Rule-based Machine Learning.** Another promising direction is the explicit integration of physical laws into the model training process via rules (Weiss & Indurkhya, 1995; Kliegr et al., 2021), constraints (Raissi et al., 2017; Cai et al., 2021), or symbolic reasoning (Yu et al., 2023; Garcez et al., 2008). This extends previous text-to-video model training frameworks that merely match videos with text, without embedding the laws of mechanics into the model architecture or loss functions. For instance, physics-informed loss functions can be introduced to penalize violations of conservation laws. This is analogous to physics-informed neural networks (PINNs) in scientific computing (Pang et al., 2019; Raissi et al., 2019; 2017; Cai et al., 2021), where differential equations (e.g., Navier–Stokes equations for fluid dynamics or simple Newtonian equations of motion) are incorporated into the loss function via automatic differentiation. Additionally, hybrid neuro-symbolic systems could offer a solution for injecting explicit physical-law reasoning into generative models (Dang-Nhu, 2020; Choi et al., 2024). Specifically, one could imagine a system where a deep generative model proposes a video sequence, and a symbolic physics engine (or differentiable simulator) evaluates and refines it. In such physics engines, if the text calls for two objects to collide, a symbolic module could compute the collision outcome using established equations of motion and enforce that outcome in the generated frames.

## 6 CONCLUSION

In this work, we have presented **T2VPhysBench**, a human-evaluated and first-principle-inspired benchmark designed to explore whether modern text-to-video models obey fundamental physical laws. Our comprehensive study reveals that, despite their impressive visual fidelity and instruction following, current models uniformly struggle to satisfy even the most basic Newtonian and conservation constraints, as well as the phenomenon principles. Moreover, performance varies markedly across law categories: conservation principles prove especially challenging compared to Newton's laws or phenomenological effects, indicating uneven modeling of different aspects of physics. Attempts to improve compliance via progressively more detailed prompt hints yield little benefit and can even degrade performance, showing that the core limitations lie beyond simple prompt design. Finally, in counterfactual tests, where models are asked to generate physically impossible scenarios, systems still produce rule-violating outputs, demonstrating reliance on pattern memorization rather than true physical reasoning. These findings highlight persistent gaps in the physical understanding of text-to-video generators. We hope T2VPhysBench will guide future efforts toward truly physics-aware video generation.

ETHIC STATEMENT

This paper does not involve human subjects, personally identifiable data, or sensitive applications. We do not foresee direct ethical risks. We follow the ICLR Code of Ethics and affirm that all aspects of this research comply with the principles of fairness, transparency, and integrity.

REPRODUCIBILITY STATEMENT

We ensure the reproducibility of our empirical findings. For all experiments, we describe the sources of the LLM models, datasets, evaluation metrics, and experiment setup in the main text. Several example prompts used are also provided to support the reproducibility of our results.

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

# Appendix

**Roadmap.** In Section A, we present the details of each evaluated model. In Section B, we show detailed video examples.

## A    IMPLEMENTATION DETAILS

We show some extra details of the selected generators in this subsection. Specifically, the implementation details for our listed models in Table 1 is presented as follows:

- **Kling** (Kling, 2024): Kling is a closed-source text-to-video model developed by Kuai and released in 2024, with four different versions: Kling 1.0, Kling 1.5, kling 1.6, and the latest, Kling 2.0. It provides both a standard and a member-only high-quality generation mode. It accepts creative parameters: increasing these settings enhances the output relevance, while reducing them fosters more creative results. It does not provide an option for camera movement. Kling is capable of producing videos lasting either 5 or 10 seconds, with flexible aspect ratios, including 16:9, 1:1, and 9:16. It also offers a prompt dictionary, AI-generated prompt hints (powered by DeepSeek), and negative prompts as optional settings. It can generate four videos in parallel from a single prompt and supports seed selection. Video generation takes approximately four minutes per sample, with a batch limit of five videos.

- **Wan 2.1** (Alibaba, 2025): Wan 2.1 is an open-source text-to-video model (WanTeam, 2025) developed by Alibaba, released in 2025. It is available in two variants: Wan 2.1 Fast and Wan 2.1 Professional. It works with multiple aspect ratios, including 16:9, 9:16, 1:1, 4:3, and 3:4. Wan 2.1 also enables extended prompt input, features an Inspiration Mode, and generates videos with sound.

- **Sora** (OpenAI, 2024): Sora is a closed-source text-to-video generator developed by OpenAI, released in 2024. It operates in a single mode and allows output in 480p, 720p, or 1080p, with aspect ratios of 16:9, 1:1, and 9:16. It supports generating 30 FPS videos lasting 5, 10, 15, or 20 seconds. A monthly fee of $20 provides access to 480p and 720p videos, each with a maximum length of 5 seconds. A $200 monthly subscription is needed for 1080p videos exceeding 5 seconds. Since most models only support 720p, the $20 subscription may be sufficient for many users. After reaching the daily limit, Sora switches to "relaxed mode," which still maintains fast video generation—around 30 seconds per video. In addition, [Sora] offers style presets and can generate four videos in parallel from the same prompt.

- **Mochi-1** (Genmo, 2024): Mochi-1 is an open-source text-to-video generator developed by Genmo and released to the public in 2024. It offers multiple modes, supporting 480p resolution, a 16:9 aspect ratio, and 5-second videos at 24FPS. It also provides random prompt suggestions and a seed function. Interestingly, when prompted to generate a video with three people, Mochi-1 often ends up creating only two. It can generate two videos in parallel, with each one taking about three minutes to process.

- **LTX Video** (HaCohen et al., 2024): LTX Video is an open-source text-to-video generator developed by Lightricks and opened to the public in 2024. It offers various preset styles and supports 768×512 (512p) resolution. It also supports aspect ratios of 16:9, 1:1, and 9:16, as well as 5-second clips at 24FPS. LTX Video allows you to specify shot type, scene location, style presets, and references, and it supports voiceover scripts. To use it, you first generate the initial scene, then generate motion for that scene.

- **Pika 2.2** (Pika, 2024): Pika 2.2 is a closed-source text-to-video model developed by Pika Labs and introduced in 2025. It offers various features, including Pikaframes, Pikaaffects, Pikascenes, Pikaaddition, and Pikawaps. Videos can be generated in 720p or 1080p resolution, with multiple aspect ratio options such as 16:9, 9:16, 1:1, 4:5, 4:3, or 5:2. You can also create clips lasting 5 or 10 seconds, with support for both negative prompts and seed inputs. I've had a great experience with Pika 2.2—the UI is easy to understand, user-friendly, and highly responsive. It generates four videos at once, each taking about 30 seconds, and lets you copy and edit prompts with a single click.

- **Dreamina** (ByteDance, 2024): Dreamina is a closed-source text-to-video model developed by Bytedance, launched in 2024. It comes in four variants: Video S2.0, Video S2.0 Pro, Video P2.0 Pro, and Video 1.2. It uses Deepseek-R1 for prompt enhancement, and provides aspect ratio choices including 16:9, 21:9, 4:3, 1:1, 3:4, and 9:16. Video S2.0, Video S2.0 Pro, and Video P2.0 Pro are able to create 5-second videos, while Video P2.0 Pro also allows for 10-second clips. Video 1.2 enables generation of videos lasting 3, 6, 9, or 12 seconds. All version operates at 24FPS.

- **Qingying** (Zhipu, 2024): Qingying serves as the commercial edition of the CogVideo family models (Hong et al., 2023; Yang et al., 2024b), which are open-source text-to-video models built by Zhipu, opened to the public in 2023 and 2024. It provides two modes for generation: Fast and Quality. Five-second videos are supported at either 60FPS or 30FPS, with aspect ratios including 16:9, 9:16, 1:1, 3:4, and 4:3. Qingying also features three advanced settings: video style, emotional atmosphere, and camera movement mode. Additionally, it supports both AI-generated sound and effects.

- **Hailuo** (MiniMax, 2025): Hailuo is a closed-source text-to-video model developed by MiniMax and introduced in 2025. It features T2V-01-Director and T2V-01 for generating videos from text. It supports 720p resolution, likely with a 16:9 aspect ratio, a 6-second duration, and 24FPS.

- **Stable Video Diffusion** (Blattmann et al., 2023a): Stable Video Diffusion is an open-source text-to-video generator developed by Stability AI, released in 2023. It provides aspect ratio choices including 16:9, 3:2, 1:1, 4:5, and 9:16. The length of generated video is 4s.

## B  VIDEO EXAMPLES

In this section, we present a wide range of video samples generated using the prompts proposed in this benchmark, as illustrated in Figures 7—30. Each figure includes results from five distinct text-to-video models, with five key frames selected from the video samples to illustrate how they change over time. These selected video instances align with all the experiments discussed in Section 4.

## Newton's First Law

Prompt: A person quickly pull out the paper pressed under the water bottle.

Figure 7: **Results of Generating Videos Following Newton's First Law**.

## Newton's First Law

Prompt: A person quickly pull out the paper pressed under the water bottle.

Figure 8: **Results of Generating Videos Following Newton's First Law**.

Newton's Second Law
Prompt: A ball rolls by, a person kicks it.

Figure 9: **Results of Generating Videos Following Newton's Second Law**.

## Newton's Second Law
Prompt: A ball rolls by, a person kicks it.

Figure 10: **Results of Generating Videos Following Newton's Second Law**.

## Newton's Third Law

Prompt: A man in a boat, paddling hard backward with an oar.

Figure 11: **Results of Generating Videos Following Newton's Third Law**.

## Newton's Third Law
Prompt: A man in a boat, paddling hard backward with an oar.

Figure 12: **Results of Generating Videos Following Newton's Third Law**.

## Law of Universal Gravitation
Prompt: A man throws a ball upward.

Figure 13: **Results of Generating Videos Following Law of Universal Gravitation**.

## Law of Universal Gravitation

Prompt: A man throws a ball upward.

Figure 14: **Results of Generating Videos Following Law of Universal Gravitation**.

## Conservation of Energy
Prompt: Two balls collide, one stops, the other moves.

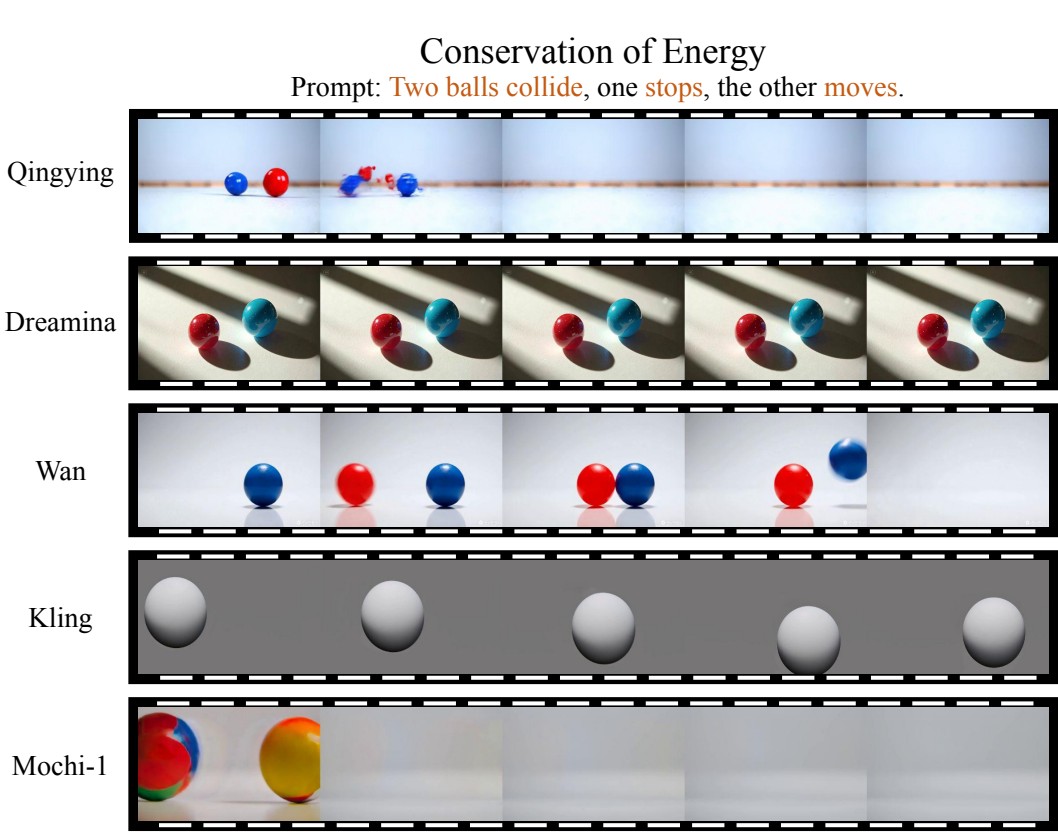

Figure 15: **Results of Generating Videos Following Conservation of Energy**.

## Conservation of Energy
Prompt: Two balls collide, one stops, the other moves.

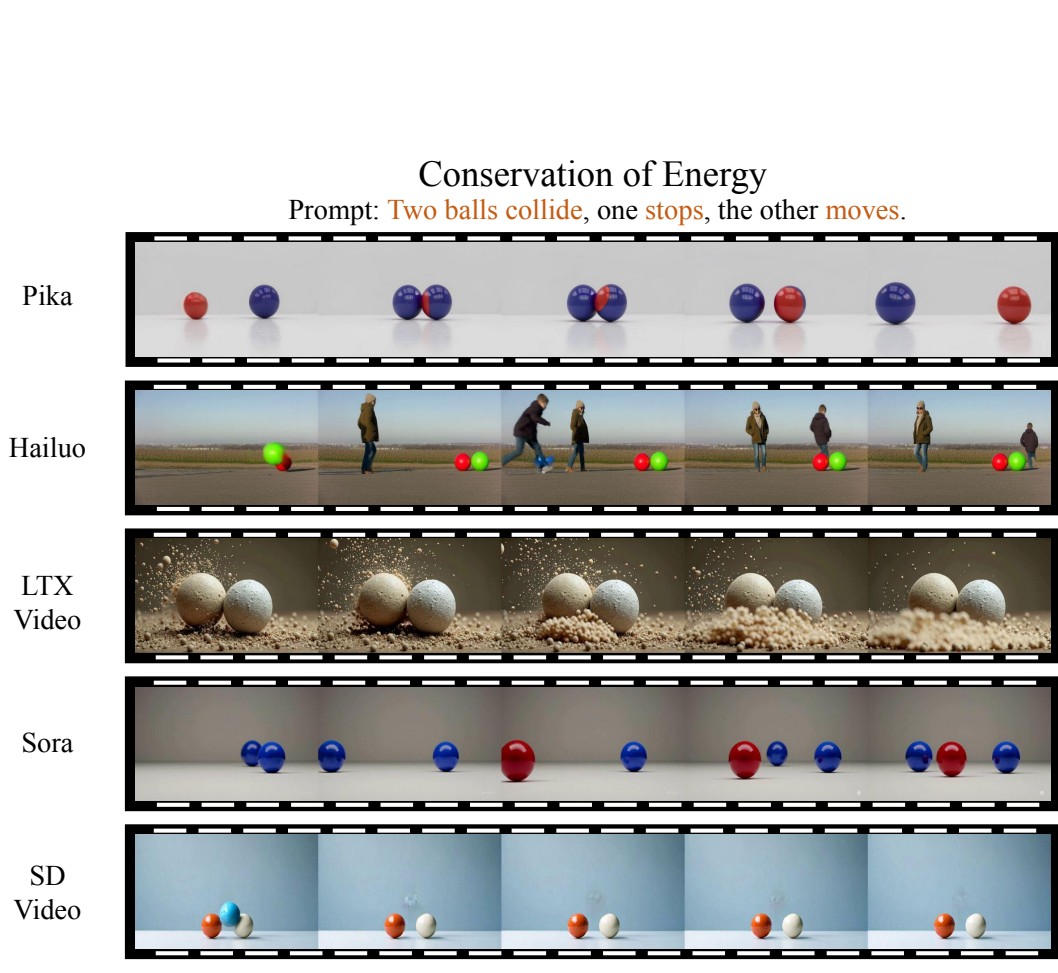

Figure 16: **Results of Generating Videos Following Conservation of Energy**.

## Conservation of Mass

Prompt: Pouring water from one cup into an empty cup, the total amount remains the same.

Figure 17: **Results of Generating Videos Following Conservation of Mass**.

## Conservation of Mass

Prompt: Pouring water from one cup into an empty cup, the total amount remains the same.

Figure 18: **Results of Generating Videos Following Conservation of Mass**.

## Conservation of Momentum

Prompt: A white pool ball hits the stationary ball, the white pool ball stops and the stationary ball flies out.

Qingying

Dreamina

Wan

Kling

Mochi-1

Figure 19: **Results of Generating Videos Following Conservation of Momentum**.

## Conservation of Momentum

Prompt: A white pool ball hits the stationary ball, the white pool ball stops and the stationary ball flies out.

Figure 20: **Results of Generating Videos Following Conservation of Momentum.**

## Conservation of Angular Momentum

Prompt: A figure skater spins with arms extended, then pulls them in to spin faster.

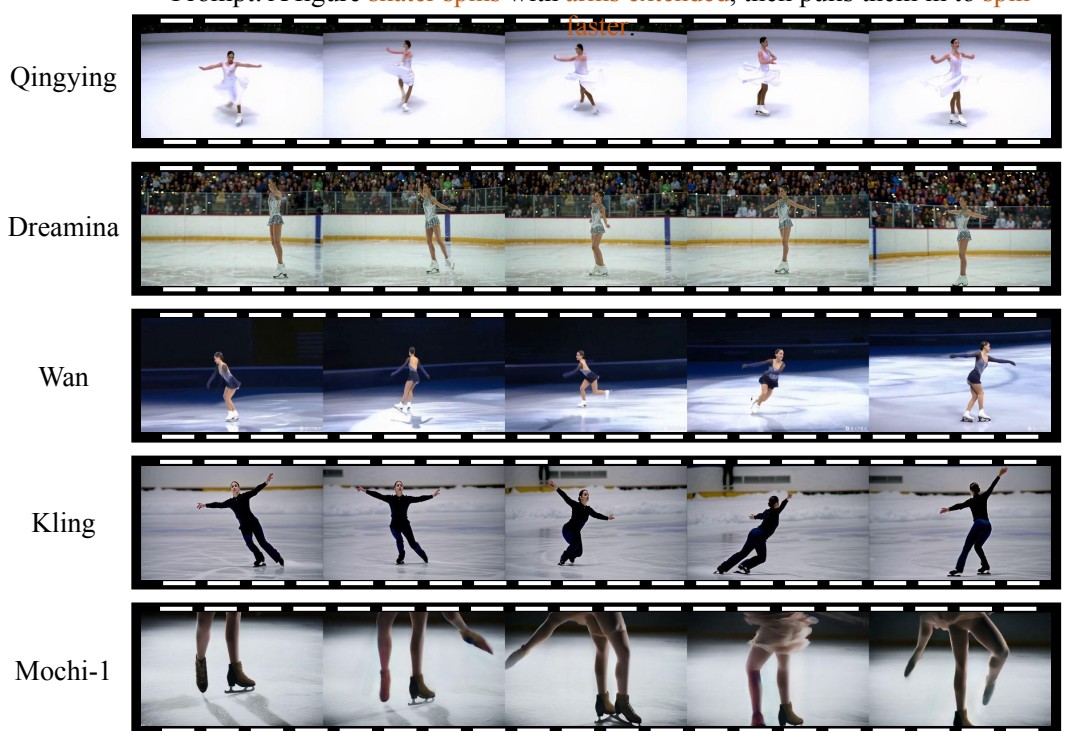

Figure 21: **Results of Generating Videos Following Conservation of Angular Momentum**.

## Conservation of Angular Momentum

Prompt: A figure skater spins with arms extended, then pulls them in to spin faster.

Figure 22: **Results of Generating Videos Following Conservation of Angular Momentum**.

## Hooke's Law
Prompt: Pressing a spring hard, the spring shortens a lot.

Figure 23: **Results of Generating Videos Following Hooke's Law**.

## Hooke's Law

Prompt: Pressing a spring hard, the spring shortens a lot.

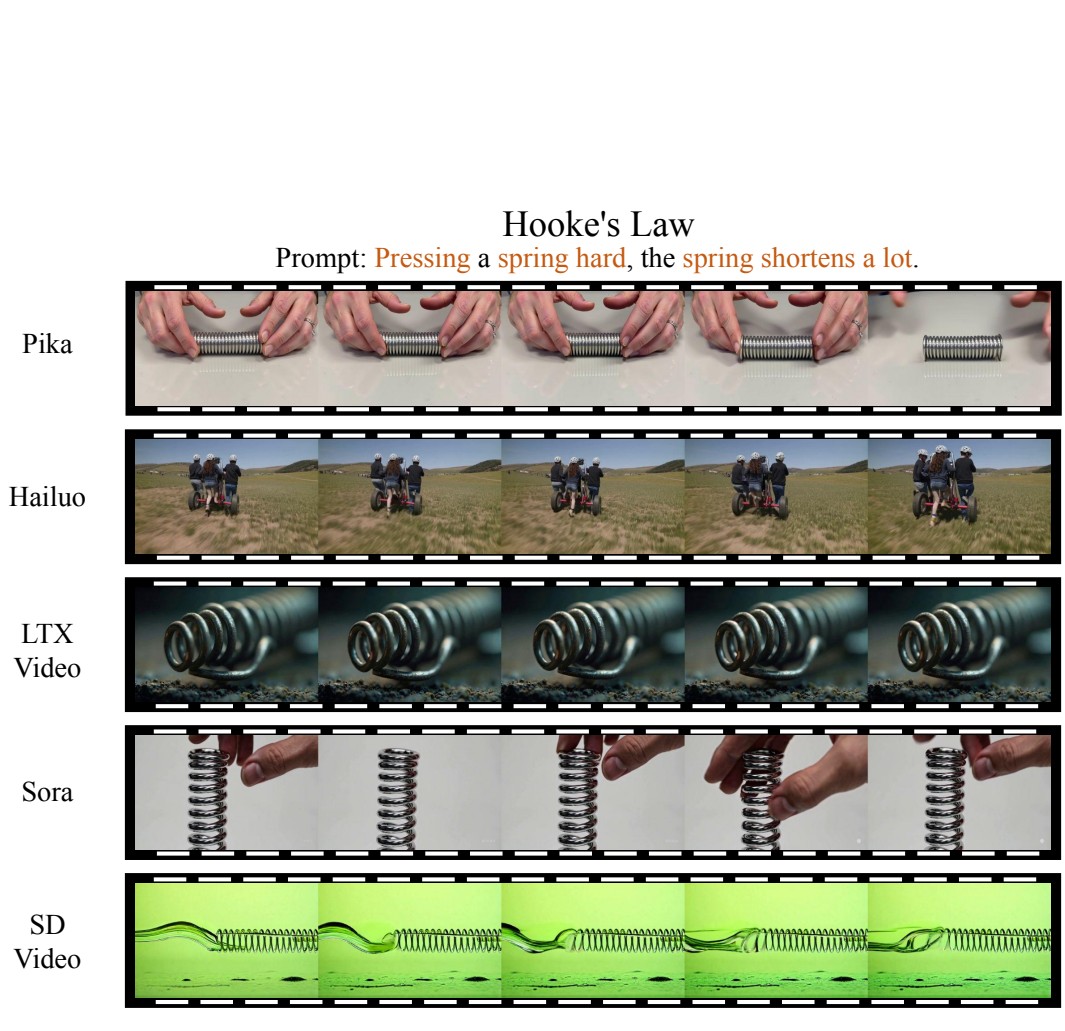

Figure 24: **Results of Generating Videos Following Hooke's Law**.

## Snell's Law

Prompt: A pencil in the water looks bent.

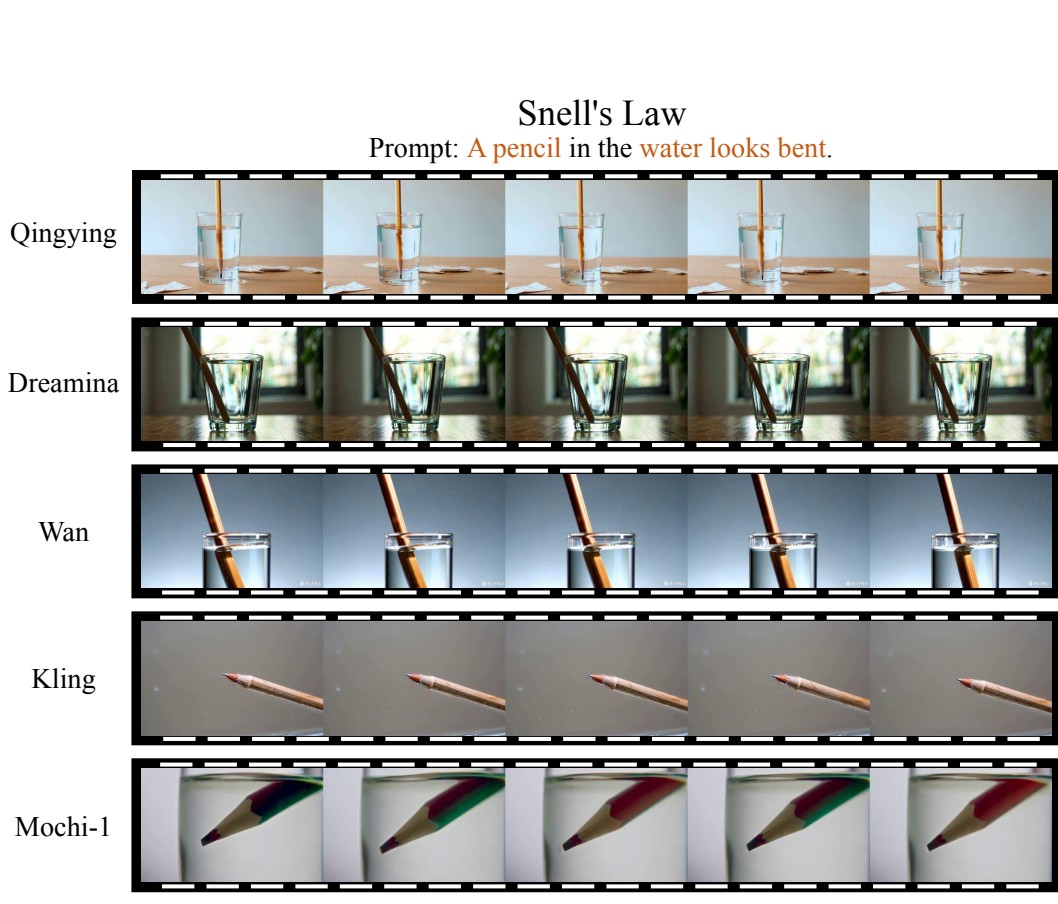

Figure 25: **Results of Generating Videos Following Snell's Law**.

## Snell's Law

Prompt: A pencil in the water looks bent.

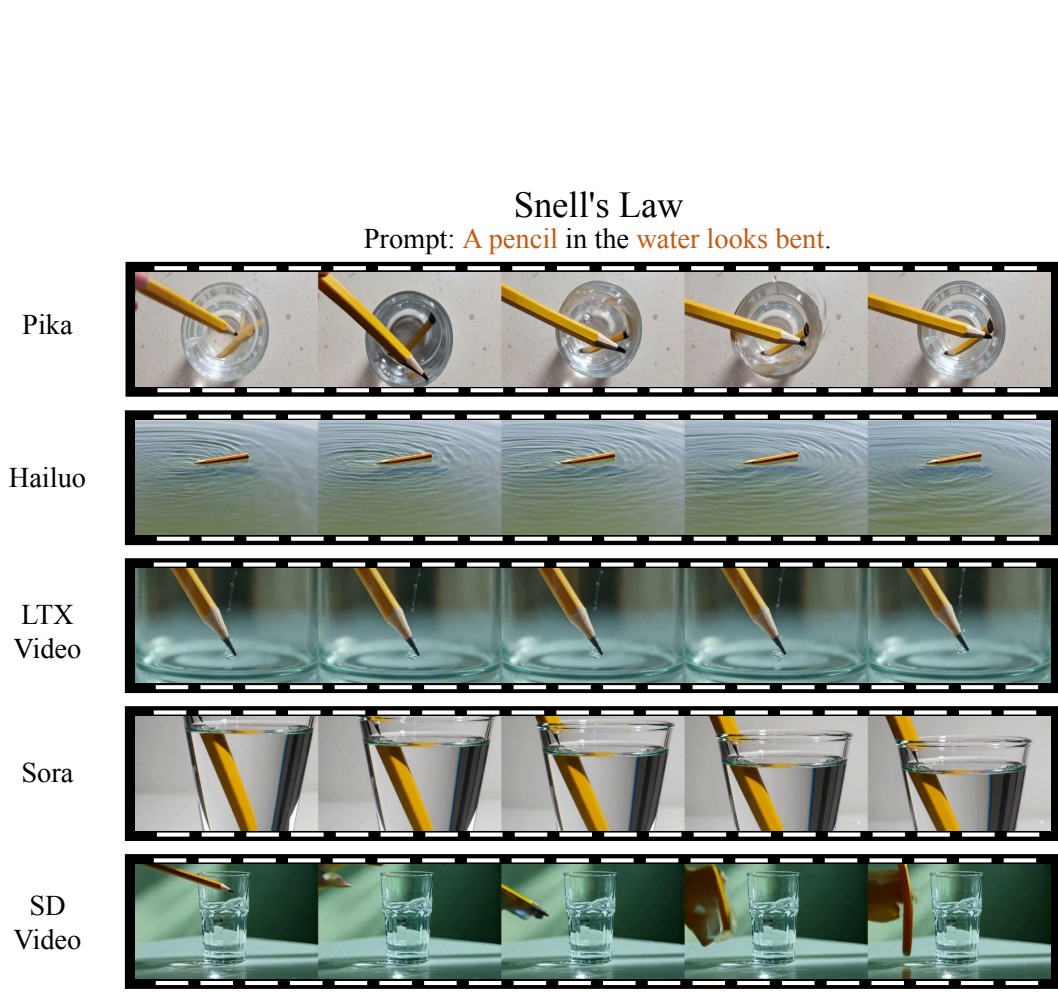

Figure 26: **Results of Generating Videos Following Snell's Law**.

## Law of Reflection

Prompt: Throw a ball diagonally at a wall and it will bounce off diagonally.

Figure 27: **Results of Generating Videos Following Law of Reflection**.

## Law of Reflection

Prompt: Throw a ball diagonally at a wall and it will bounce off diagonally.

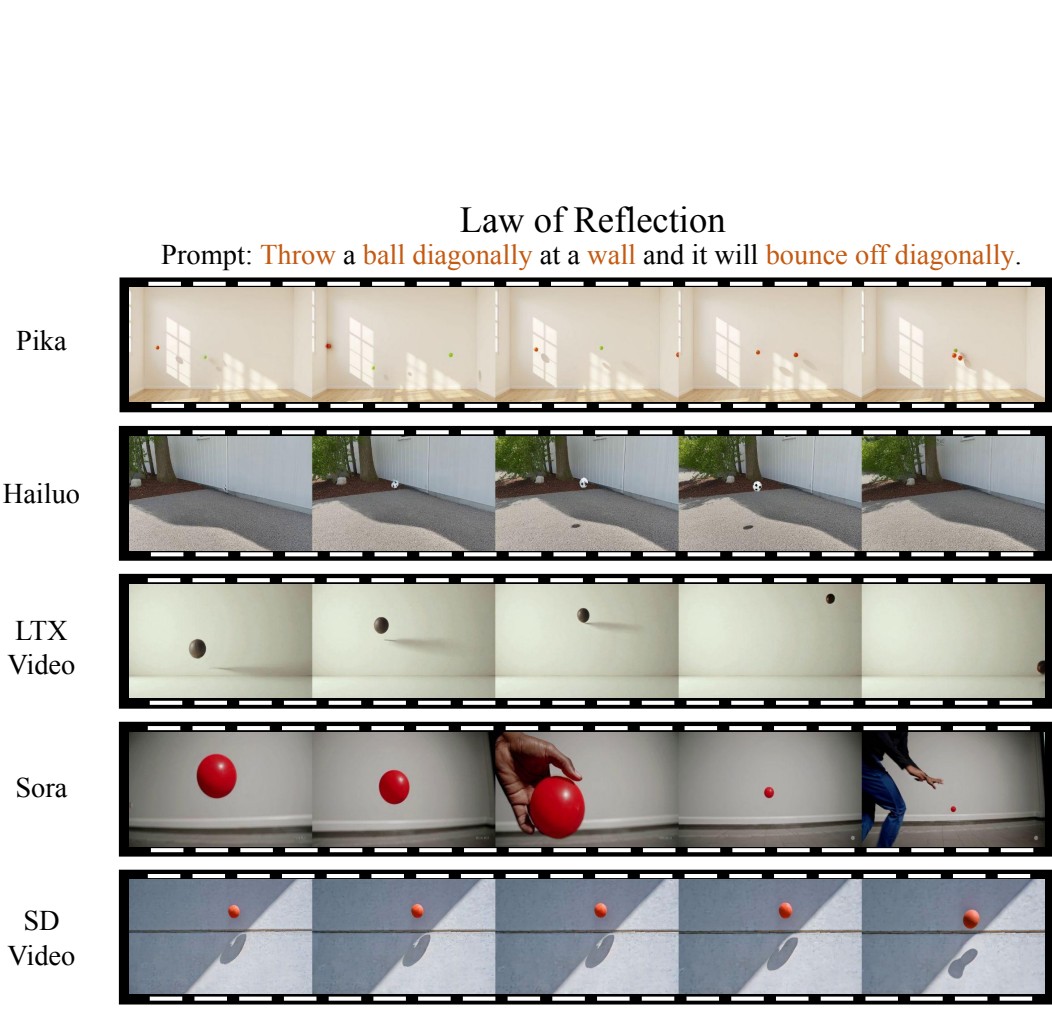

Figure 28: **Results of Generating Videos Following Law of Reflection**.

## Bernoulli's Principle

Prompt: A spinning ball bends in flight.

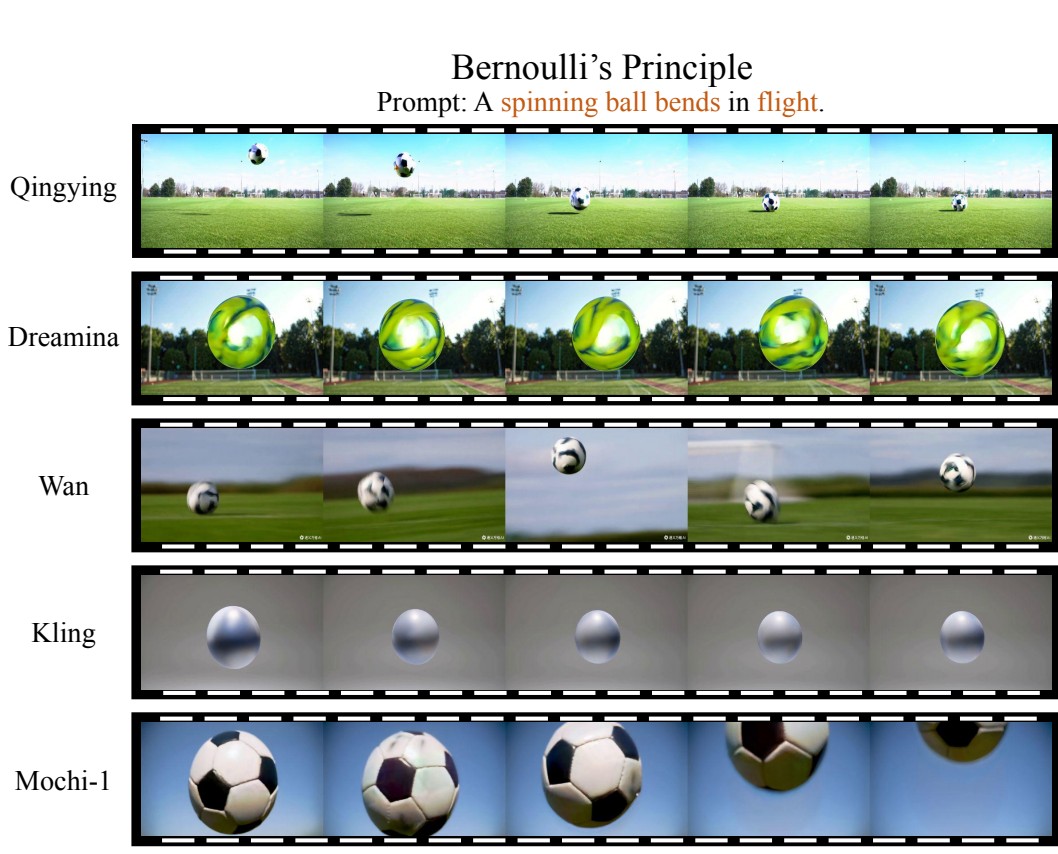

Figure 29: **Results of Generating Videos Following Bernoulli's Principle**.

## Bernoulli's Principle

Prompt: A spinning ball bends in flight.

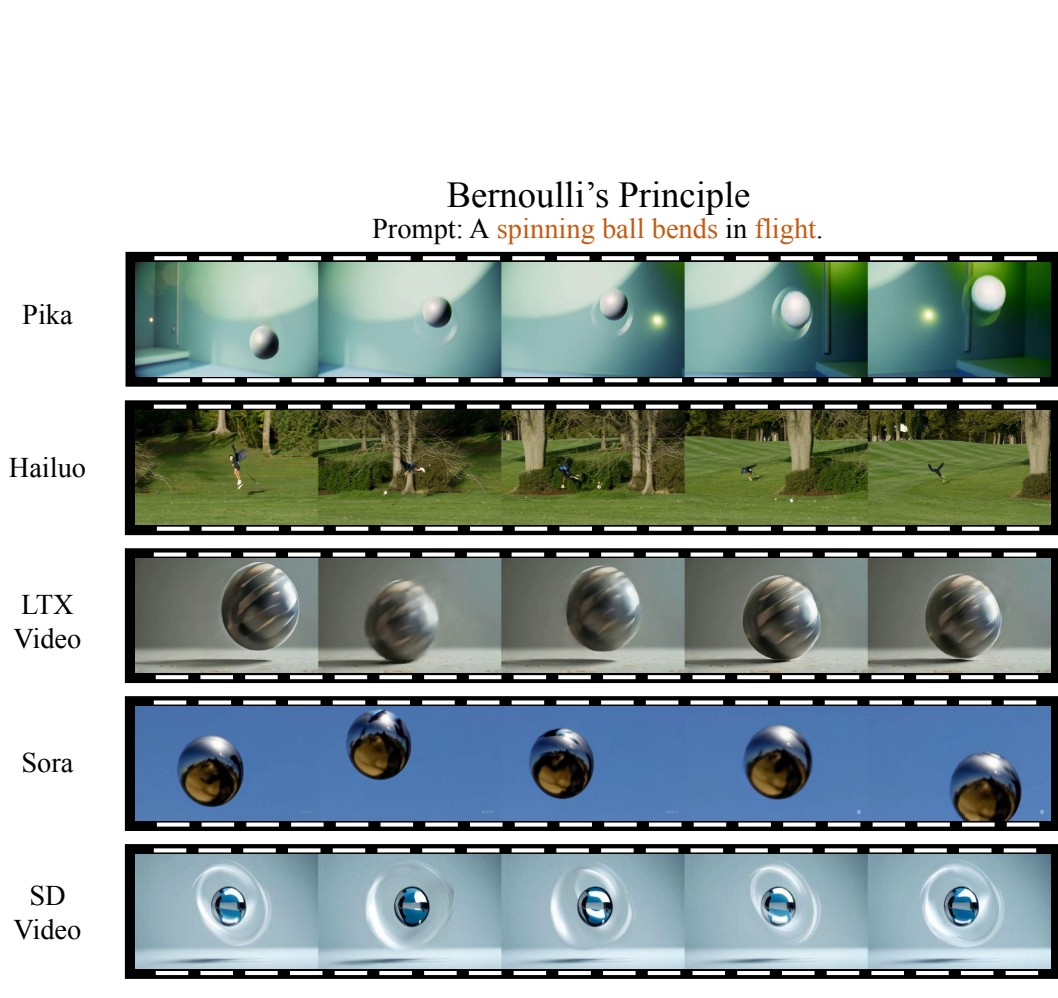

Figure 30: **Results of Generating Videos Following Bernoulli's Principle**.

## LLM Usage Disclosure

LLMs were used only to polish language, such as grammar and wording. These models did not contribute to idea creation or writing, and the authors take full responsibility for this paper's content.

