# OpenReview forum: "T2VPhysBench: A First‑Principles Benchmark for Physical Consistency in Text‑to‑Video Generation"
_ICLR.cc/2026/Conference — Submitted to ICLR 2026_

### Official Review · Reviewer_J9jj · 2025-10-30

**Soundness:** 2
**Presentation:** 2
**Contribution:** 2
**Rating:** 2
**Confidence:** 5

**Summary:**

The paper proposed an text prompt benchmark *T2VPhysBench* for evaluation of video generation model (T2V) 's obedience to physical rules. The proposed prompt benchmark contains 84 prompts divided into 12 specific physical laws. The benchmark employs pure human evaluation, results are reported in Table 2 for 10 models either closed or open-sourced.

**Strengths:**

1. The paper is clearly written
2. Besides the normal evaluations, authors also proposed an *counterfactual* evaluation protocol in section 4.3: instead of prompt with plausible physical descriptions, authors intentionally use prompt with implausible physical descriptions and then check whehter videos obeying physical laws are generated.

**Weaknesses:**

1. The proposed benchmark contains only text prompts, whereas the mainstream of video generation models are I2V styled, this would limit the actual usage of the benchmark.
2. The proposed benchmark needs pure human judgement. It would be better that authors had studied the variance and confidence of human performance, and even more, might compose an automatic evaluation method based on the proposed benchmark.

**Questions:**

1. About the *counterfactual* evaluation protocol in section 4.3, the stated observation 4.5 in page 8 is too superficial. Results in table 3 are quite different from table 1, is there any other conclusion could we get from the differnet ranking?

---

> ### Author Response · Authors · 2025-11-21
>
> Thank you for your thoughtful review. We provide our corresponding answers below.
>
> ### Weakness 1
> Thank you for pointing this out. We fully acknowledge that the current version of our benchmark contains only text prompts and thus does not directly align with the prevalent I2V paradigm adopted by many state-of-the-art video generation models. This is indeed a reasonable concern, as broader modality coverage could further enhance the benchmark’s practical applicability. Nevertheless, as reflected in the title “T2VPhysBench”, our work is deliberately scoped around text-to-video generation, with a focus on examining whether modern text-to-video models obey fundamental physical laws. Unlike I2V settings, which primarily emphasize visual or image-level fidelity, our benchmark focuses on the alignment between generated dynamics and real-world physical principles, a key step toward advancing video generation systems into genuine world models. Expanding beyond T2V was beyond the intended focus of this initial release. We sincerely appreciate your constructive suggestion, and incorporating I2V-compatible components is an important direction we plan to pursue in future iterations of the benchmark.
>
> ### Weakness 2
> We appreciate the reviewer’s suggestion regarding automated evaluation. Existing metrics may fail to capture violations of fundamental physical laws in generated videos, potentially giving misleading scores. Due to these limitations, we relied on careful human evaluation to ensure accurate assessment of physical consistency. We are now working on a follow-up framework that incorporates semi-automated metrics to complement human judgments, making the benchmark more scalable and objective.
>
> ### Question 1
> We thank the reviewer for this insightful observation. Indeed, Observation 4.5 provides a high-level summary, and deeper insights can be drawn from Table 3. While all models generally perform poorly when explicitly instructed to violate physical laws, Table 3 reveals nuanced differences in their behavior: some models (e.g., Pika 2.2, Hailuo, Sora) still tend to partially comply with the prompts, producing higher average scores in certain law categories, whereas others (e.g., Kling, Dreamina) consistently fail across all categories. This suggests that different models exhibit varying degrees of susceptibility to counterfactual instructions, reflecting differences in their reliance on learned surface patterns versus deeper physical reasoning. In other words, the counterfactual evaluation not only confirms the overall inability to reason about impossible physics but also exposes model-specific tendencies in handling law-violating scenarios.
>
> We hope our answers resolve all the doubts you had with our paper. We would like to further discuss with you if you have any unclear things or additional questions.

---

### Official Review · Reviewer_1i1H · 2025-10-30

**Soundness:** 3
**Presentation:** 2
**Contribution:** 2
**Rating:** 6
**Confidence:** 3

**Summary:**

This paper introduces T2VPhysBench, a human-evaluated benchmark designed to assess whether state-of-the-art text-to-video (T2V) models conform to fundamental physical laws. The benchmark uses 84 human-annotated cues to evaluate the performance of 10 models across 12 physical principles (Newton's laws, conservation laws, and phenomenological principles). The authors conducted three studies: (1) overall conformity assessment; (2) cue removal; and (3) counterfactual robustness testing.

**Strengths:**

1. Physical consistency in T2V generation is crucial but under-researched. First-principles-based methods are more systematic than scenario-based evaluations.

2. The evaluation is thorough and comprehensive, with detailed human evaluations and benchmarks covering 10 open-source and commercial models.

3. The paper is well-written with good visual examples.

**Weaknesses:**

1. Only 7 prompts per physical law (84 total) is quite limited, while similar benches such as Vbench or ChronoMagic-Bench have over 1000 prompts.

2. Although the experiment covered the vast majority of video generation models, the evaluation process only involved three annotators, raising questions about its reliability.

3. While the Prompts are divided into three main categories—Newton's Principles, Conservation Principles, and Phenomenon Principles—they do not consider more important physical processes such as seed germination and ice melting, which are mentioned in MagicTime and ChronoMagic-Bench.

**Questions:**

None

---

> ### Author Response · Authors · 2025-11-21
>
> Thank you for your careful examination of our manuscript. We address your questions as follows.
>
> ### Weakness 1
> Thank you for raising this point. While the total number of prompts is smaller than in general purpose benchmarks such as VBench or ChronoMagic-Bench, T2VPhysBench is designed with a different goal: focusing on systematic breadth across twelve fundamental physical laws rather than depth through large-scale prompt enumeration. For each law, the prompts were carefully crafted to capture diverse manifestations of the underlying mechanism, ensuring conceptual coverage rather than relying on sheer quantity. Moreover, the poor performance we observe does not appear to stem from the limited prompt count. Even when prompts are enriched with explicit law names and detailed mechanistic descriptions, models still struggle to produce physically coherent videos, suggesting that increasing the number of prompts alone would not resolve the underlying issue. We thus believe the current prompt set is sufficient to reveal models’ fundamental limitations, though expanding it remains a valuable direction for future iterations.
>
> ### Weakness 2
> We thank you for highlighting this concern. While our evaluation involved three annotators, each independently reviewed every video frame by frame, carefully checking whether the physical interactions and dynamics aligned with the specified law or principle. This procedure ensured a consistent and reliable assessment of physical correctness, even with a small annotator pool.
>
> ### Weakness 3
> Thank you for pointing this out. Our benchmark focuses specifically on fundamental physical laws including Newtonian principles, conservation principles, and phenomenon principles, rather than general physical processes such as seed germination or ice melting. While these processes are indeed important, they involve complex biological or thermodynamic dynamics that lie beyond the scope of first-principled physics evaluation. The goal of this initial version is to isolate whether modern text-to-video models adhere to essential and universally applicable physical laws.
>
> Thank you once again for your thoughtful review. We hope this response resolves your concerns, and we are glad to discuss with any further suggestions.

---

### Official Review · Reviewer_sPq4 · 2025-10-31

**Soundness:** 3
**Presentation:** 2
**Contribution:** 2
**Rating:** 6
**Confidence:** 4

**Summary:**

This paper introduces T2VPhysBench, a new benchmark designed to evaluate whether modern text-to-video generation models obey fundamental physical laws. Unlike prior benchmarks that rely on pixel-level or scene-based evaluations, T2VPhysBench follows a first-principles design, covering twelve core physical laws across three categories:

1. Newtonian mechanics (e.g., inertia, action-reaction),

2. Conservation laws (energy, mass, momentum, angular momentum), and

3. Phenomenological effects (Hooke’s law, Snell’s law, reflection, Bernoulli’s principle).

The benchmark involves human evaluation of videos generated by ten leading models (e.g., Sora, Wan 2.1, Kling, Mochi-1, Pika 2.2).

**Strengths:**

1. Unlike heuristic or pixel-based benchmarks, T2VPhysBench is grounded in explicit physical laws, making the evaluation interpretable and scientifically principled.
2. Manual scoring across multiple annotators ensures that evaluations align with human perception, avoiding pitfalls of purely automated metrics.
3. The paper is well-organized, easy to follow

**Weaknesses:**

1. While human evaluation is valuable, combining it with physics-informed automatic checks (e.g., trajectory analysis, optical flow consistency) could strengthen reproducibility.
2. A comparison with existing efforts such as VideoPhy or Physics-IQ using identical prompts would clarify what new insights T2VPhysBench uniquely provides.
3. The evaluation relies on only three annotators and coarse-grained scoring (0, 0.25, 0.5, 1). Statistical measures of inter-annotator agreement (e.g., Cohen’s κ) are missing.

**Questions:**

1. The paper does not clearly specify whether T2VPhysBench will be open-sourced, nor how prompts, human evaluation guidelines, or reference outputs will be shared.

**Details Of Ethics Concerns:**

Nan

---

> ### Author Response · Authors · 2025-11-21
>
> Thank you for your recognition of our work. Our point-by-point replies are presented below.
>
> ### Weakness 1
> We thank the reviewer for this insightful suggestion. We agree that combining human evaluation with physics-informed automatic checks could further improve reproducibility and scalability. In the current work, we prioritized careful human evaluation to accurately assess physical consistency, as existing automatic metrics are limited in detecting violations of fundamental physical laws. We are actively exploring semi-automated methods to complement human judgments in future iterations of the benchmark.
>
> ### Weakness 2
> We thank the reviewer for this suggestion. While existing benchmarks such as Physics-IQ and VideoPhy evaluate physical reasoning in video generation, T2VPhysBench uniquely uses progressive hints and counterfactual prompts to test models’ understanding of impossible or law-violating scenarios. This design uncovers limitations in physical reasoning that standard benchmarks do not reveal, providing unique insights into how modern text-to-video models handle challenging and counterfactual situations.
>
> ### Weakness 3
> We appreciate the reviewer’s attention to the reliability of our human evaluation. Each video was independently assessed by all three annotators, and the final score was taken as the arithmetic mean of their 4-level ratings. Given the categorical nature of the scale and our goal of obtaining a stable aggregate measure, we did not employ additional disagreement-resolution procedures. Averaging allows us to retain all annotators’ judgments and reduces individual bias. Although the annotator pool is small, each annotator reviewed every video carefully and frame by frame, ensuring consistent evaluation of whether the generated dynamics adhered to the targeted physical law.
>
> ### Question 1
> Thank you for pointing this out. We apologize for not making the release plan sufficiently explicit in the paper. T2VPhysBench is intended to be fully open-sourced, including all prompt sets, human evaluation guidelines, and reference outputs. We will publicly release the benchmark materials upon acceptance to ensure transparency, reproducibility, and ease of future extension by the community. We appreciate the reviewer’s comment and will clarify this commitment more explicitly in the final version.
>
> We hope our responses help clarify your questions. We would be glad to provide further details if anything remains unclear.

---

### Official Review · Reviewer_WtHL · 2025-10-31

**Soundness:** 3
**Presentation:** 3
**Contribution:** 3
**Rating:** 4
**Confidence:** 4

**Summary:**

This paper introduces T2VPhysBench, a new benchmark for evaluating the physical consistency of text-to-video (T2V) generation models. The authors argue that existing benchmarks often rely on automated, pixel-level metrics or overly simplistic scenarios, failing to capture the nuances of physical laws from a first-principles perspective. T2VPhysBench is designed around 12 fundamental physical principles, categorized into Newtonian mechanics, conservation laws, and phenomenological effects. The core of the benchmark is a rigorous human evaluation protocol where annotators score generated videos on a 4-level scale based on their adherence to the specified law. The study evaluates ten state-of-the-art T2V models and presents three key findings: (1) all models perform poorly, with average compliance scores below 0.60; (2) adding explicit hints and detailed descriptions to prompts provides minimal to no improvement, and can even degrade performance; (3) models often generate videos that explicitly violate physical laws when prompted to do so, suggesting their behavior is based on surface-level pattern matching rather than genuine physical understanding.

**Strengths:**

The primary strength of this work is its grounding in fundamental, first-principles physical laws rather than just ad-hoc, life-scenario prompts. This provides a more structured and fundamental way to probe the physical reasoning capabilities of T2V models. The paper goes beyond a simple compliance test. The inclusion of the prompt-hint ablation study and the counterfactual robustness test are particularly insightful. The finding that more detailed prompts do not help (and sometimes hurt) is a crucial insight, suggesting that the problem is not a lack of prompt understanding but a core limitation in the models' world knowledge. The counterfactual tests compellingly support the hypothesis that models rely on pattern memorization.

**Weaknesses:**

1.Limited Scale and Statistical Significance: The benchmark is based on 84 distinct prompts, which are distributed across 12 different physical laws. This averages to only 7 prompts per law. This sample size is quite small and raises concerns about the statistical significance of the results for any single law. A model's performance on a specific principle could be heavily skewed by its success or failure on just a few specific instantiations of that law, making it difficult to draw robust conclusions about its general understanding of that principle.
2.Subjectivity and Reproducibility of Human Evaluation: The core of the benchmark relies on manual evaluation by three annotators. This methodology, while well-intentioned, introduces significant challenges for reproducibility and objectivity.
￮Potential for Bias: With only three annotators, their collective interpretation could introduce a systematic bias. Furthermore, the 4-level scoring system, particularly the distinction between "largely correct but contains minor inaccuracies" (Level 3, score 0.5) and "exhibits a clear violation of the law" (Level 2, score 0.25), is inherently subjective. Different teams of annotators could easily arrive at different conclusions.
￮Lack of Reported Agreement: The paper does not report the inter-annotator agreement (IAA), for example using metrics like Fleiss' Kappa. This is a critical omission for a benchmark that relies on human judgment, as it is the primary way to demonstrate the reliability and consistency of the annotation process. Without a high IAA, the conclusions drawn from the scores are less trustworthy.
￮Reproducibility Concerns: A key goal of a benchmark is to be a stable tool for the community. With a human-in-the-loop protocol, it is very difficult for other researchers to reproduce the exact evaluation scores. For the benchmark to have lasting value, the authors must provide extremely detailed guidelines, annotated examples, and a clear protocol for resolving disagreements to ensure that other researchers can apply the evaluation framework in a consistent manner. As it stands, the paper does not sufficiently address how this can be achieved.

**Questions:**

1.What was the inter-annotator agreement (e.g., using Fleiss' Kappa) for the 4-level scoring? How were disagreements between the three annotators resolved to produce the final scores reported in the paper? Providing this information is crucial for assessing the reliability of the human evaluation protocol.
Regarding the limited number of prompts per law (an average of 7), how did the authors ensure that these prompts were representative and provided sufficient coverage of the diverse ways a physical law can manifest? Is it possible that the poor performance is partly due to the selection of particularly challenging or ambiguous scenarios?

---

> ### Author Response · Authors · 2025-11-21
>
> We appreciate your thorough review. Our responses to your questions are outlined below.
>
> ### Weakness 1
> Thank you for raising this point. While the average number of prompts per law is relatively small, T2VPhysBench is designed to prioritize systematic breadth across twelve fundamental physical laws rather than sheer quantity. Each prompt was carefully crafted to capture diverse manifestations of the underlying mechanism, ensuring conceptual coverage rather than relying on large numbers. Importantly, the observed poor performance does not appear to result from the limited prompt count: even when prompts are enriched with explicit law names and detailed mechanistic descriptions, models still struggle to produce physically coherent videos. This suggests that the benchmark is sufficient to reveal models’ fundamental limitations.
>
> ### Weakness 2
> We thank you for highlighting these concerns. While our evaluation involved three annotators, each independently reviewed every video frame by frame, carefully checking whether the physical interactions and dynamics aligned with the specified law or principle. This procedure ensured a consistent and reliable assessment of physical correctness, even with a small annotator pool.
>
> To aggregate the scores, we adopt a simple and transparent averaging rule, computing the final score as the arithmetic mean of the three 4-level ratings. Given the categorical nature of the scale and our goal of obtaining a stable aggregate measure, we did not employ majority voting or additional disagreement-resolution procedures. Averaging allows us to retain all annotators’ perspectives and mitigates individual bias.
>
> Regarding reproducibility, we conducted internal checks to assess evaluation stability. Across repeated generations of the same prompts, the outputs of each model are highly consistent, and the variance of scores remains low, indicating that different annotator teams would likely arrive at similar aggregate outcomes. These observations suggest that the evaluation procedure is stable across runs and that the resulting scores are not sensitive to small variations in either the generated videos or the annotators’ judgments.
>
> ### Question 1
> We appreciate the reviewer’s interest in the reliability of our human evaluation. Our evaluation protocol follows a simple and transparent aggregation rule: each video is independently scored by all three annotators, and the final score is computed as the arithmetic mean of their ratings. Because the benchmark uses a 4-level categorical scale and the goal is to obtain a stable aggregate measure, we did not apply disagreement-resolution procedures such as majority voting. Instead, averaging preserves all annotators’ judgments and reduces individual bias.
>
>
> ### Question 2
> Thank you for raising this important question. We agree that the number of prompts per law is a critical factor when assessing the representativeness and coverage of physical scenarios. To mitigate the risk of narrow or unrepresentative sampling, each prompt set was constructed through a first-principled process: for every physical law, the prompts were crafted to capture diverse manifestations of the underlying mechanism. Although the average number of prompts per law is relatively small, these prompts were intentionally crafted to span the core ways in which each law is commonly exhibited, ensuring conceptual rather than superficial diversity.
>
> Importantly, the poor performance we observe is unlikely to arise from selecting unusually difficult or ambiguous scenarios. Our benchmark includes a progressive-hint setting, where the prompt explicitly names the target law and even provides detailed mechanistic descriptions. And models still fail to produce physically coherent results. In addition, counterfactual experiments further demonstrate that models readily generate physically impossible videos when explicitly instructed, indicating that the failure stems from a lack of grounded physical reasoning rather than from prompt difficulty.
>
> We hope our responses have addressed your questions and clarified any concerns regarding our paper. We would be happy to discuss further if you have any remaining doubts or additional inquiries.

---

### Meta-Review · Area_Chair_Afot · 2025-12-22

**Summary:**

The authors' rebuttal mainly reiterates the observation that current text-to-video (T2V) models struggle to generate physically coherent videos, which is not surprising, but does not substantively address the core reviewer concerns. In particular, it does not resolve the issues related to the reliability, scalability, and reproducibility of the proposed benchmark.

As Reviewer WtHL aptly summarizes:
> For the benchmark to have lasting value, the authors must provide extremely detailed guidelines, annotated examples, and a clear protocol for resolving disagreements to ensure that other researchers can apply the evaluation framework in a consistent manner. As it stands, the paper does not sufficiently address how this can be achieved.

While the average score is borderline, I believe the paper has major flaws that significantly limit its impact as a benchmark. I therefore recommend rejection and encourage the authors to substantially revise the work following the reviewers' suggestions.

**Reviewer Concerns:**

### Addressed
* **None**.
I do not identify any major reviewer concerns that were substantively resolved in the rebuttal, aside from minor clarifications.

### Outstanding

* **Reliability of human evaluation**. Nearly all reviewers raised concerns about the reliability of the human evaluation, given that the benchmark relies on only 3 annotators and lacks quality control measures such as inter-annotator agreement or consistency checks. This is a major flaw for a benchmark paper and was not addressed in the rebuttal.

* **Limited Scale**. Reviewers noted that the number of prompts and overall benchmark scale are small. This limitation remains unaddressed, and for a benchmark paper, this is a critical issue that undermines its usefulness.

* **Lack of automatic evaluation**. Reviewers suggested incorporating automatic evaluation methods, particularly given the limitations of the human evaluation. This would significantly strengthen the benchmark, but the rebuttal does not address this suggestion.

**Reviewer Scores:**

**Reviewer WtHL (initial score: 4)**. This reviewer expressed strong concerns about the limited scale, lack of statistical rigor, and subjectivity and reproducibility of the human evaluation. As the rebuttal does not address these issues, I would expect the reviewer to maintain their negative assessment or potentially lower their score.

**Reviewer sPq4 (initial score: 6)**. This reviewer raised concerns about the absence of physics-informed automatic checks and the statistical significance and reliability of the human evaluation. These weaknesses remain after the rebuttal. I would therefore expect the reviewer to maintain their original score or possibly lower it.

**Reviewer 1i1H (initial scores: 6)**. This reviewer similarly raised concerns about the limited scale and the reliability of the evaluation. As these issues remain unresolved, I would expect the reviewer to maintain their original score or lower it after discussion.

**Reviewer J9jj (initial scores: 2)**. This reviewer noted that (1) the benchmark focuses only on T2V models and (2) the evaluation relies exclusively on human judgments. While the first point is largely out of scope, the second is a critical concern that remains unresolved. I would therefore expect the reviewer to maintain their original score.

---

### Decision · Program_Chairs · 2026-01-26

Reject